

# 1 Stratospheric ozone depletion inside the volcanic plume shortly after the 2 2022 Hunga Tonga eruption

Yunqian Zhu[1,2,3], Robert W. Portmann[1], Douglas Kinnison[4], Owen Brian Toon[3,5], Luis Millán[6],
Jun Zhang[4], Holger Vömel[7], Simone Tilmes[4], Charles G. Bardeen[4], Xinyue Wang[4], Stephanie
Evan[8], William J. Randel[4], Karen H. Rosenlof[1]
1. NOAA, Chemical Sciences Laboratory
2. Cooperative Institute for Research in Environmental Sciences, University of Colorado
10         Boulder
3. Laboratory for Atmospheric and Space Physics, University of Colorado Boulder
4. NCAR, Atmospheric Chemistry Observations and Modeling Laboratory
5. Department of Atmospheric and Oceanic Sciences, University of Colorado Boulder
6. Jet Propulsion Laboratory, California Institute of Technology, 4800 Oak Grove Drive,
15        Pasadena, CA 91109, USA
7. NCAR, Earth Observing Laboratory
8. Laboratoire de l'Atmosphère et des Cyclones (LACy, UMR8105, CNRS, Université de La
18        Réunion, Météo-France), Saint-Denis, France

Corresponding author: Yunqian Zhu (yunqian.zhu@noaa.gov)
**Abstract**
In-plume ozone depletion was observed for about ten days by Microwave Limb Sounder
(Aura/MLS) right after the January 2022 Hunga Tonga-Hunga Ha'apai (HTHH) eruption. This
work analyzes the dynamic and chemical causes of this ozone depletion. The results show that
the large water injection (~150 Tg) from the HTHH eruption, with ~0.0013 Tg injection of ClO
(or ~0.0009 Tg of HCl), causes ozone loss due to strongly enhanced HOx and ClOx cycles and
their interactions. Aside from the gas phase chemistry, the heterogeneous reaction rate for
$HOCl+HCl\rightarrow Cl_2+H_2O$ increases to $10^4$ cm$^{-3}$sec$^{-1}$ and is a major cause of chlorine activation,
making this event unique compared with the springtime polar ozone depletion where
$HCl+ClONO_2$ is more important. The large water injection causes relative humidity over ice to
increase to 70% - 100%, decreases the $H_2SO_4/H_2O$ binary solution weight percent to 35%
compared with the 70% ambient value, and decreases the plume temperature by 2-6 K. These
changes lead to high heterogeneous reaction rates. Plume lofting of ozone-poor air is evident
during the first two days after the eruption, but ozone concentrations quickly recover because its
chemical lifetime is short at 20 hPa. With such a large seawater injection, we expect that ~5 Tg
Cl was lifted into the stratosphere by the HTHH eruption in the form of NaCl, but only ~0.02%
of that remained as active chlorine in the stratosphere. lightning NOx changes are probably not
the reason for the HTHH initial in-plume $O_3$ loss.
**Key points:**
● HOCl is identified as playing a large role in the in-plume chlorine balance and
heterogeneous processes, making this event unique compared with the ozone hole where
$HCl+ClONO_2$ is more important.
● The HTHH eruption enhanced the HOx/ClOx cycles and their interactions, which caused
in-plume $O_3$ depletion.





● The injection of Cl, H$_2$O, and lightning NOx modified the ambient chemistry.

### 1. Introduction

Stratospheric ozone concentrations change after volcanic eruptions for a variety of reasons.
Enhanced polar ozone depletion occurs after large or medium volcanic eruptions [*Hofmann and*
*Oltmans*, 1993; *Portmann et al.*, 1996; *Solomon et al.*, 2016] since heterogeneous reactions on
volcanically enhanced sulfate aerosols result in amplified anthropogenic ClOx and BrOx induced
ozone loss. *Tie and Brasseur* [1995] demonstrated that mid- and high latitude O$_3$ changes after a
volcanic eruption largely depend on chlorine loading. For the pre-industrial era and in the
absence of anthropogenic halogens in the stratosphere, O$_3$ would slightly increase in the middle
atmosphere after a large volcanic eruption resulting from the suppression of NOx-catalyzed
destruction by heterogenous creation of HNO$_3$ on volcanic aerosols. After the 1991 Pinatubo
eruption, the radiative heating caused by volcanic aerosols perturbed the local temperature and
circulation, which lifted the ozone layer and caused equatorial ozone depletion [*Kinnison et al.*,
1994]. *Wang et al.* [2022] reported that, in the case of the Hunga-Tonga eruption, mid-latitude
ozone reduction was primarily caused by anomalous upwelling. Enhanced water can also change
O$_3$. In the lower most stratosphere, H$_2$O injection through deep convection or tropopause cirrus
clouds could change the catalytic chlorine/bromine free-radical chemistry and shift the total
available inorganic chlorine towards the catalytically active free-radical form, ClO [*Solomon et*
*al.*, 1997; *Anderson et al.*, 2012].
*Evan et al.* [2023, submitted] report observations of decreased O$_3$ and HCl, and increased
ClO in the first week following the HTHH eruption at 20 hPa. Here we use the
CESM2(WACCM6) model [*Zhu et al.*, 2022] to analyze the dynamic and chemical contributors
to this initial in-plume ozone depletion. A lofting plume can bring ozone-poor tropospheric air
into the stratosphere and cause in-plume low ozone values compared with the surrounding
stratospheric air [*Yu et al.*, 2019]. For a submarine volcanic eruption, the in-plume air
composition is not only impacted by tropospheric air, but also by the seawater, and volcanic
gases (including H$_2$O, CO$_2$, SO$_2$, HCl, HF, H$_2$S, S$_2$, H$_2$, CO, and SiF$_4$.), and volcanic minerals.
For the HTHH initial plume, besides high H$_2$O and high SO$_2$, Microwave Limb Sounder (MLS)
observations indicate the in-plume air carried high CO (**Figure A1**), relatively low ozone, and
high ClO, compared with the surrounding air. We constrain the initial plume chemical
compounds based on observational data from MLS; then analyze how stratospheric chemistry
changes the plume composition. We will answer the following scientific questions:
1. What are the initial conditions in the volcanic plume?
2. What are the main causes of in-plume ozone depletion?
3. How do volcanic injections impact heterogeneous reactions that cause chlorine activation
in the plume?

### 2. Observational data description and model setup

The MLS instrument onboard the EOS Aura satellite was launched into a near-polar sun-
synchronous orbit in 2004. This work uses MLS version 4 for O$_3$, ClO, temperature, and CO data
during the first ten days after the eruption as recommended by *Millán et al.* [2022]. The vertical
resolution of these MLS products is typically around 3-5 km in the stratosphere. All data used
here were screened using the methodology indicated in *Livesey et al.* [2022]. We use the MLS
H$_2$O data to identify the plume location and define it as regions with water vapor larger than 10
ppmv.





93   *Vömel et al.* [2022] provide water vapor radiosonde measurements during the first three
94 global circumnavigations of the plume. Here we calculate the relative humidity relative to ice
95 (RHi) and compare the observed values with the simulated values.
96   We use the 70-layer Whole Atmosphere Community Climate Model (WACCM) model as
97 described in *Zhu et al.* [2022], injecting $SO_2$ (0.42 Tg) and $H_2O$ (150 Tg). The model's vertical
98 resolution is about 1 km in the stratosphere. The model atmosphere is nudged to GEOS5
99 meteorological analysis [*Rienecker et al.,* 2008] until January 14, one day before the eruption
100 day. After January 15, we run the model freely with a fully interactive atmosphere and ocean for
101 ten days.
102   We constrain the simulated volcanic aerosol, $H_2O$, and chlorine by comparing to
103 observations during the first ten days after the eruption. *Zhu et al.* [2022] show that the simulated
104 aerosol backscatter coefficient agrees with the CALIPSO observations on January 17. The
105 simulated $H_2O$ agrees with MLS [*Millán et al.*, 2022; *Zhu et al.*, 2022] from February 1 to April
106 1, 2022. Here, we compare the simulated $H_2O$ with the radiosonde observations of humidity
107 [*Vömel et al.*, 2022] during the first week. **Figure 1** shows the RHi on January 18 and January 19
108 observed by the radiosonde and from nearby simulated model output. Both the observations and
109 simulations show relative humidity between 70% to 100%. The radiosonde observations have a
110 much higher vertical resolution than the model. Therefore, they show multiple layers of water
111 enhancement, while the model only shows one.

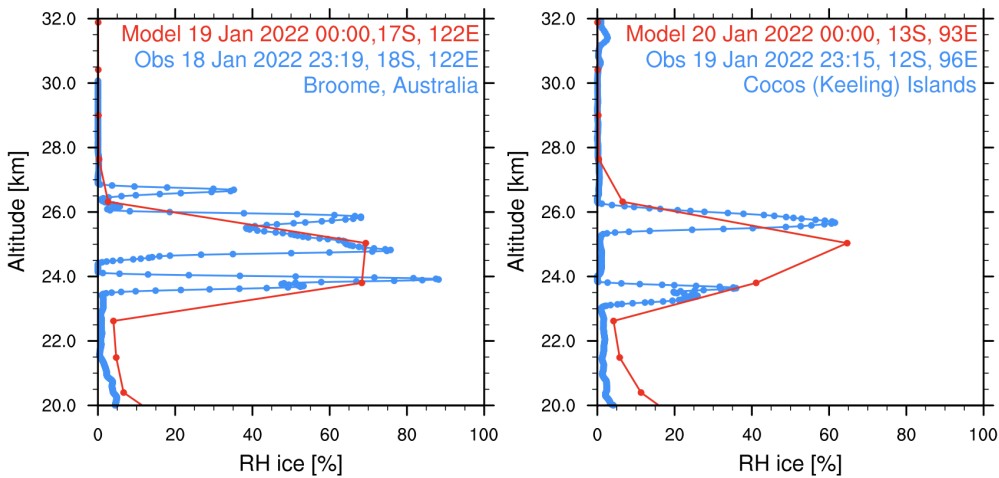

113 **Figure 1.** Relative humidity with respect to ice saturation vapor pressure from radiosondes (blue)
114 [*Vömel et al.*, 2022] and simulation (red). The profiles are picked at nearby locations. Note the
115 observations are about 45 minutes earlier in time than the simulations, which places them on a
116 different day.

118   We constrain the chlorine injection using MLS ClO observations at 20 hPa. **Figure 2a**
119 shows ClO from the MLS observations and the model simulations at 20 hPa from January 18 to
120 January 24. MLS values are selected from locations where water vapor is larger than 10 ppmv,
121 indicating these values are inside the volcanic plume. **Figures 2b** and **2c** show the simulated
122 daytime ClO for one plume location for each day. The dates are marked next to each plume.
123 MLS observations show elevated ClO, about 5 to 10 times higher than the ambient values
124 (**Figure 2a**). If we only inject $SO_2$ and $H_2O$ (The H2O_SO2 case defined in Table 1), we get a





ClO amount about twice as large as the background (**Figure 2b**), which is much lower than
observed. The change of ClO indicates that $H_2O$ alters the Cly partitioning. To match the
observed values, we need to inject 0.0013 Tg of ClO (**Figure 2c**). This is equivalent to injecting
~0.0009 Tg of HCl (**Figure A2**). In our simulations, injecting ClO and HCl does not lead to
different HOCl (**Figure A3**), ClO, and $O_3$ levels after January 15, indicating the balancing of
ClO and HCl inside the HTHH plume happens very quickly. Unfortunately, the HOCl retrieval
from MLS is not suitable for scientific use at this pressure level, so we cannot validate it. We
choose the ClO injection case in our following analysis. Note that the MLS ClO vertical
resolution is ~2 km near 20 hPa, which is coarser than the model vertical resolution (~1 km at 20
hPa).

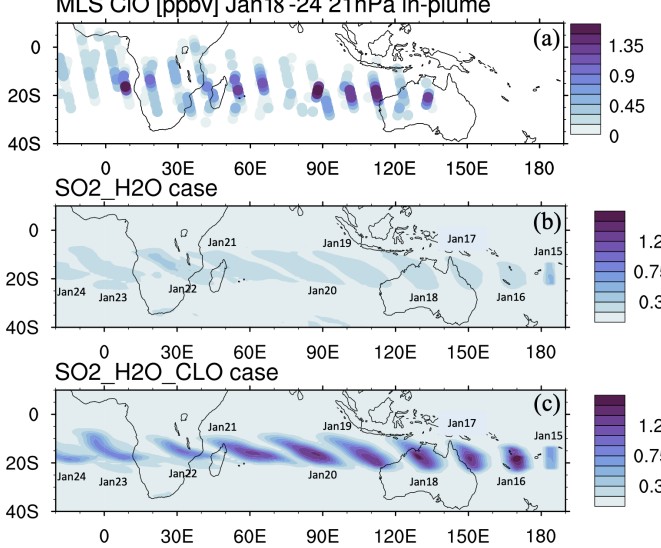

**Figure 2. a)** MLS in-plume ClO observations during the first 10 days after the eruption. "In-
plume" is defined as the area with water vapor mixing ratios larger than 10 ppmv. MLS in-plume
ClO data is not recommended for scientific use until January 18, 2022. **b)** and **c)** Simulated 10-
day evolution of in-plume ClO in the SO2_H2O and SO2_H2O_CLO case. The modeled ClO
concentrations are only taken during daytime each day (either 6 UTC or 12 UTC).
To investigate the $O_3$ decrease and its related chemical evolution during the first 10 days,
we conduct several simulations as described in **Table 1**.
**Table 1.** Model cases description.

| Name | Description |
| --- | --- |
| Nonvolc | No injection of volcanic $H_2O$ and $SO_2$. |
| H2O_SO2 | $H_2O$ and $SO_2$ injection profile follows Zhu et al. [2022]. |
| H2O_SO2_ClO | Besides $H_2O$ and $SO_2$, injection of 0.00013 Tg of ClO. ClO injection profile is proportional to $H_2O$ injection. |



| H2O_SO2_ClO_nohet | Same setting as H2O_SO2_ClO, but turn off the heterogeneous chemical reactions for $HCl+HOCl$, $ClONO_2+H_2O$, and $ClONO_2+HCl$ |
| --- | --- |
| SO2_ClO | $SO_2$ injection profile follows Zhu et al. (2022). No water injected. Injection of 0.00013 Tg of ClO using the same profile as H2O_SO2_ClO. |
| lowO3 | Reduce the $O_3$ to 75% of its original value at 20 hPa. |
| H2O_SO2_lowO3 | $H_2O$ and $SO_2$ injection, plus reducing $O_3$ to 75%. |
| H2O_SO2_ClO_lowO3 | $H_2O$, $SO_2$ and ClO injection, plus reducing $O_3$ to 75%. |
| H2O_SO2_NO | Injection of 0.003 Tg of NO in addition to $H_2O$ and $SO_2$. |

## 3. Results

     *Evan et al.* [2023] show the HTHH in-plume ozone depletion at 20 hPa lasts at least ten days after the HTHH eruption, which they attribute to the heterogeneous chlorine activation on humidified volcanic aerosols. Here we analyze the contributions to this initial in-plume $O_3$ depletion considering three processes: 1) increasing $H_2O$ injection may enhance the HOx catalytic cycle and HOx/ClOx interactions; 2) increasing ClO during the injection phase may deplete ozone due to both heterogeneous reactions and gas phase reactions; 3) the rising plume from the troposphere may carry ozone-poor tropospheric air into the stratosphere.

     **Figure 3a** shows the MLS observed in-plume ozone depletion at 20 hPa. Because the plume is spatially small during the initial days, MLS tracks do not capture the maximum plume perturbation every day. MLS measures low ozone concentrations of 4.8 ppmv on January 17, 4.6 ppmv on January 20, and 5.1 ppmv on January 24. These are ozone anomalies of about 1.7 ppmv, 1.9 ppmv, and 1.4 ppmv, respectively. The anomalies are calculated using the background average values in this area (6.5 ppmv) subtracting the low ozone values. Note that any interpretation of these $O_3$ anomalies needs to consider the coarse MLS vertical resolution (~3 km). **Figure 3b** shows the simulated $O_3$ in the H2O_SO2 case using one model time step each day that occurs near local noon. **Figure 3b** shows evident $O_3$ reduction, but less than observed, because of the water injection, which accelerates the HOx catalytic cycle. **Figure 3c** shows that once we inject ClO on top of the massive water injection, $O_3$ loss is significantly enhanced and is close to the observations after January 18. **Figure 3d** uses the same injection as **Figure 3c** but with heterogeneous reactions (i.e., $HCl+HOCl$, $ClONO_2+H_2O$, and $ClONO_2+HCl$) turned off. The difference between **Figure 3d** and **Figure 3c** is caused by heterogeneous reactions, which usually only happen in the stratospheric polar springtime where they cause the Antarctic ozone hole and Arctic ozone depletion. Heterogeneous reactions become important, despite the high non-polar temperatures because of the massive quantity of water injected. The heterogeneous reaction rate is strongly related to the relative humidity. Usually, during the polar night, the relative humidity is higher (RHi 60%-100%) than in the non-polar stratosphere because of the low temperature (<195 K). Here, the water injection increases the relative humidity (**Figure 4c**). Enhanced water causes the weight percent of $H_2SO_4$ of the sulfuric acid aerosol to decrease from 70% to 35% (**Figure 4b**). The massive water injection also causes the in-plume temperature to drop about 2 to 6 K (**Figure 4f**). All these factors (temperature decrease, relative humidity increase, and particle $H_2SO_4$ dilution) can increase the three heterogeneous reaction probabilities ($HCl+HOCl$, $ClONO_2+H_2O$, and $ClONO_2+HCl$). As shown in **Figure 5**, when the water vapor




amount is near the climatological value of 6 ppmv, the heterogeneous reaction probability
reaches $10^{-2}$ to $10^{-1}$ when the temperature is ~190 K. Meanwhile, the reaction probability is
similar for temperatures of 215 K when the water vapor is ~600 ppmv in the simulations, as was
the case for the HTHH plume during the week following the eruption. COSMIC-2 radio
occultation observed even higher water vapor during the first week: the maximum values over
Januray 20-22 are ~1000-2000 ppmv [*Randel et al.,* 2023]. Also, because the in-plume and the
out-of-plume chemical concentrations are different, we apply both conditions (solid and dashed
lines) to show how the different HCl, HOCl, and $ClONO_2$ conditions alter the HCl+HOCl and
$ClONO_2$+HCl reactions probabilities by one order of magnitude. Volcanic sulfur injection also
increases the sulfate surface area density (**Figure 4a**) that provides extra surfaces for
heterogeneous reactions.
Comparing **Figure 3b** and **3c** with MLS observations, we can see that the chemical
reactions do not explain the $O_3$ loss during the first three days of the eruption (January 15 -
January 17, low $O_3$ near 160˚E in MLS observation). This discrepancy suggests that the plume
contains some ozone-poor tropospheric air after the injection into the stratosphere. We ran three
cases with initial low ozone. For the low $O_3$ case (**Figure 3e**), we inject only ozone-poor air
without volcanic $H_2O$ and $SO_2$. It shows low $O_3$ as observed during the first couple of days, but
ozone recovers quickly because the $O_3$ chemical lifetime is short at 20 hPa inside the plume
(**Figure A4**). The H2O_SO2_lowO3 case (**Figure 3f**) shows ozone loss similar to the
observation in the first six or seven days. By adding the ClO and initial ozone-poor air (**Figure
3g**), we obtain persistent low $O_3$ values that agree with the observational lowest values better
than the other cases (**Figure 6a**). Compared with **Figure 3b**, **Figure 3d** has slightly more ozone
depletion, indicating that the extra chlorine injection impacts $O_3$ even without heterogeneous
chemistry. However, without including the high amounts of injected water, the additional ClO
alone cannot deplete ozone much (**Figure 3h**).

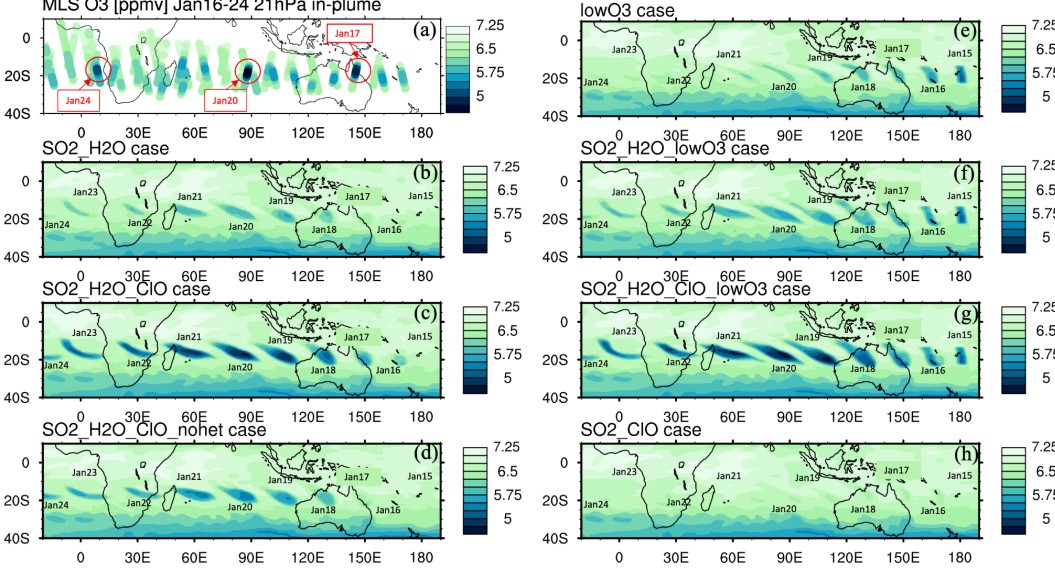




**Figure 3.** **a)** MLS in-plume $O_3$ observation during the first ten days. The locations and days
with low $O_3$ values used in **Figure 6** are marked with circles. **b-h)** Simulated 10-day evolution of
in-plume $O_3$ in seven model cases with various injections of $SO_2$, $H_2O$, ClO, and low initial $O_3$.

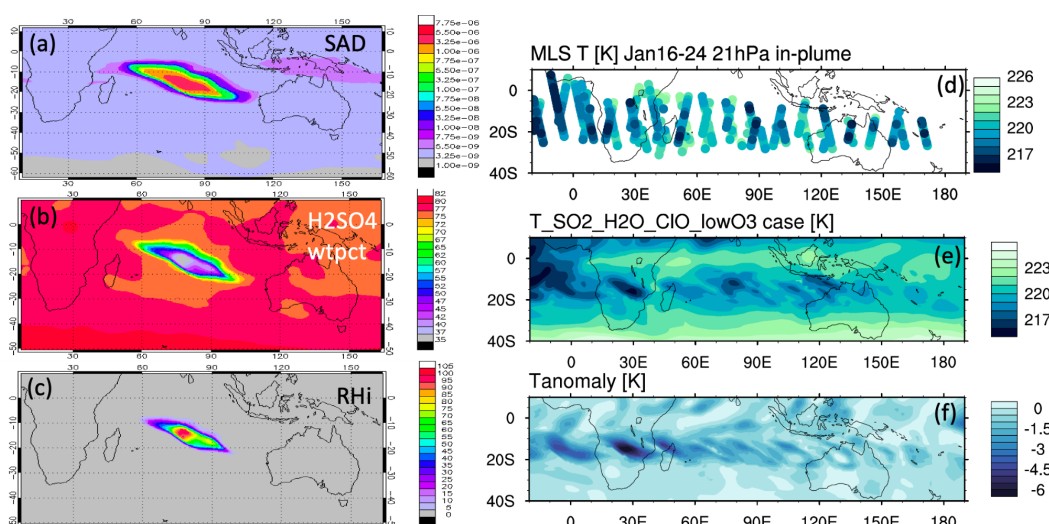

**Figure 4. a)** Simulated surface area density, **b)** simulated $H_2SO_4/H_2O$ weight percent and **c)**
relative humidity on January 20 at 20 hPa. **d)** Temperature evolution during the first ten days at
20 hPa from MLS, **e)** simulated temperature evolution in the SO2_H2O_ClO_lowO3 case; **f)**
temperature difference between the SO2_H2O_ClO_lowO3 case and the Nonvolc case.

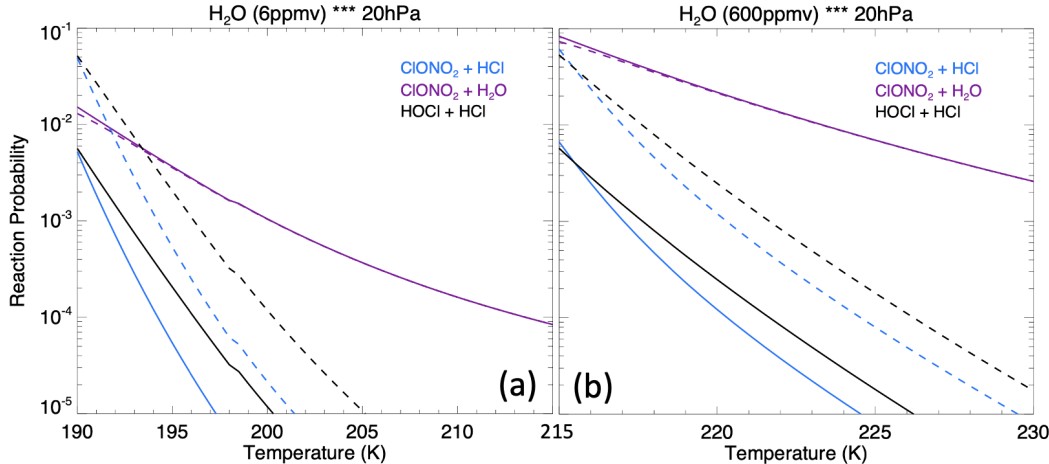

**Figure 5.** The heterogeneous reaction probability for three reactions on sulfate surfaces
($ClONO_2$+HCl, $ClONO_2$+ $H_2O$ and HOCl+HCl) as a function of water vapor assuming 0.4 μm
particle size at 20 hPa. Panel **a)** assumes 6 ppmv of ambient water vapor and panel **b)** assumes
600 ppmv of ambient water vapor. The solid lines use the out-of-plume chemical concentration
on January 20: 1.0 ppbv of HCl, 0.03 ppbv of HOCl, and 0.5 ppbv of $ClONO_2$; the dashed lines



use the in-plume chemical concentration: 0.1 ppbv of HCl, 1.0 ppbv of HOCl, and 0.05 ppbv of
ClONO$_2$. These values are based on the simulation output.

**Figure 6** shows the O$_3$ anomaly evolution from several model cases (**a**) and percentage
contributions to the total ozone loss (**b, c**). The model case with all injections (initial low O$_3$,
high H$_2$O, and high ClO) agrees well with MLS observations on the three days with the lowest
O$_3$ values (**Figure 6a**). In **Figure 6b** and **6c**, the black bars represent the contribution from the
low O$_3$ injection, which is significant during the first couple of days but diminishes quickly.
From these percentage values, we conclude that the low O$_3$ carried in the plume lofting cannot be
the reason for the low O$_3$ values after 3 days. Chemistry is the main reason that this O$_3$ depletion
lasts so long.
There are two ways to look at the chemical contributors to ozone loss based on our model
runs. The first is to separate the contributors due to various injections (**Figure 6c**): H$_2$O injection
accounts for about 30-40% of the ozone loss most of the time (blue) and ClO injection accounts
for 50% of the ozone loss most of the time (red). However, we cannot simply attribute the largest
contribution to the ClO injection, because if we only inject ClO, it does not produce much ozone
depletion (**Figure 6a,** magenta). It is the ClOx/HOx interactions that accelerate O$_3$ depletion.
A second way to look at the causes for ozone loss is to separate the contributions from
the gas-phase chemistry and the heterogeneous chemistry (**Figure 6b**). The model run with the
H$_2$O and ClO injections, but without the heterogeneous chemistry shows that the gas-phase
chemistry (yellow bars) account for more than 47% of the ozone loss from January 18 - 24.
Heterogeneous chemistry (green bars) destroys about 30% of the ozone. Hence, both
heterogeneous chemistry and gas-phase chemistry are important for O$_3$ depletion. Once we turn
off the heterogeneous chemistry, the partitioning between active chlorine and chlorine in the
reservoirs is changed. The order in which the processes are accounted for can affect the resulting
breakdown. Thus, we cannot simply say that gas phase chemistry contributions are larger than
heterogeneous chemistry. Both are clearly significant.

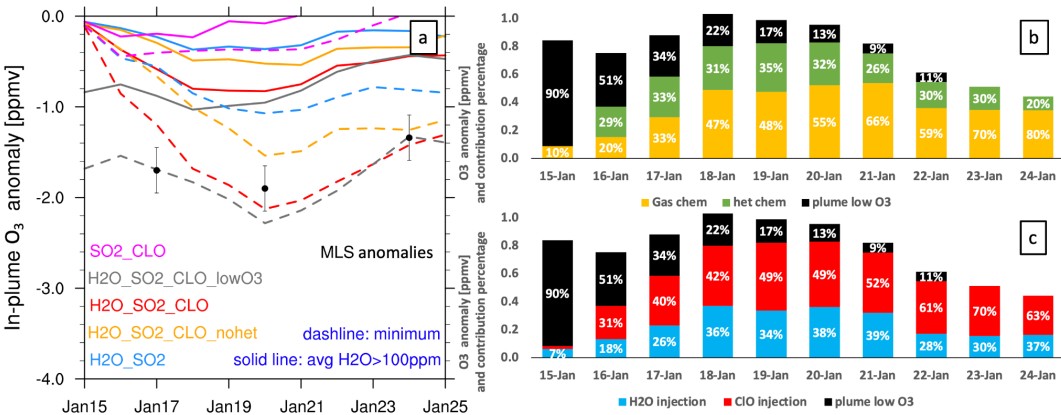

**Figure 6. a)** O$_3$ anomaly in different model cases. The solid lines are the average O$_3$ anomaly at
20 hPa on each day near local noon where water vapor is larger than 100 ppmv. 100 ppmv here
is suggested by *Evan et al.* [2023], who found that O$_3$ anomalies are not significant for a 10
ppmv but significant for a 100 ppmv threshold. The dashed lines are the simulated maximum O$_3$
anomaly on each day at 20 hPa. The black dots show the three days during which MLS measures



the lowest $O_3$ values (explained in **Figure 3a**). **b)** The percentage contributions to ozone loss
from gas phase chemistry (orange) (H2O_SO2_CLO_nohet), heterogeneous chemistry (green,
H2O_SO2_CLO minus H2O_SO2_CLO_nohet), and low $O_3$ air carried into the stratosphere
(black, H2O_SO2_CLO_lowO3 minus H2O_SO2_CLO). **c)** The percentage contributions to
ozone loss from $H_2O$ injection (blue, H2O_SO2 minus Nonvolc), ClO injection (red,
H2O_SO2_CLO minus H2O_SO2), and low $O_3$ air carried into the stratosphere (black,
H2O_SO2_CLO_lowO3 minus H2O_SO2_CLO).

To better understand which reactions are critical in the HTHH plume, we investigate the
simulated reaction rates related to HOx and chlorine compounds (**Figure 7**). These reactions
reflect how the water and ClO injections strengthen the in-plume HOx/ClOx interactions,
chlorine activation, and the relative importance of each heterogeneous reaction rate. The
WACCM model uses the methods developed by *Shi et al.* [2001] for heterogeneous reaction rate
calculations. **Figure 7a** shows the HOx cycle inside and outside the water plume on Januray 20,
daytime, at 20 hPa. The $HO_2+O_3$ reaction rate increases from $5x10^4$ to $5x10^5$ cm$^{-3}$sec$^{-1}$; OH+O
increases from $2x10^4$ to $10^5$ cm$^{-3}$sec$^{-1}$; $HO_2$+O increases from $2x10^3$ to $10^4$ cm$^{-3}$sec$^{-1}$. In addition,
the extra HOx plays a large role in chlorine activation. **Figure 7b** shows the chlorine compound
reactions inside the HTHH initial plume. The HOCl photolysis rate increases from $5x10^3$
cm$^{-3}$sec$^{-1}$ outside the plume to $10^5$ cm$^{-3}$sec$^{-1}$ inside the plume, which is the dominant process
causing the increase in chlorine activation to Cl. The HOCl concentration remains high due to
the enhanced ClOx/HOx interaction (i.e., $ClO+HO_2 \rightarrow HOCl+O_2$ reaction), as well as the increase
of the heterogeneous reaction rate of $ClONO_2+H_2O$ from $10^{-2}$ to $4x10^4$ cm$^{-3}$sec$^{-1}$. The large
amounts of HOCl also make the heterogeneous reaction of HOCl+HCl faster than the
$ClONO_2$+HCl reaction, while the latter reaction is known as the major reaction contributing to
the chlorine activation that contributes to the polar ozone depletion. **Figure A5** shows the uptake
coefficient for the three heterogeneous reactions HCl+HOCl, $ClONO_2+H_2O$, and $ClONO_2$+HCl
on January 20. The reaction rate of $ClONO_2$+HCl is increased to $10^{-2}$ cm$^{-3}$sec$^{-1}$ compared with
the background value of $10^{-10}$ cm$^{-3}$sec$^{-1}$. This value is even higher than *Evan et al.* [2023]
suggested, who estimate that enhanced water increases the uptake coefficient of $ClONO_2$+HCl to
$10^{-4}$ cm$^{-3}$sec$^{-1}$. The reaction probability of HCl+HOCl and $ClONO_2+H_2O$ increases to $10^{-2}$ cm$^{-}$
$^3$sec$^{-1}$. Furthermore, inside the plume, the reactions that convert Cl back to HCl are slower than
their activation rate.
**Figure 7c** shows another process significantly altered by the water plume. $HO_2$+NO is
usually not an important process for $O_3$ production in the stratosphere (more important in the
troposphere). The reaction rate increases from $3x10^5$ cm$^{-3}$sec$^{-1}$ outside the plume to $7x10^5$ cm$^{-}$
$^3$sec$^{-1}$ inside the plume.
**Figure 8** shows the contributions to Cly
$(Cl+ClO+2Cl_2+2Cl_2O_2+OClO+HOCl+ClONO_2+HCl+BrCl)$ and the percentage of each
compound inside and outside the plume. Outside the plume, HCl and $ClONO_2$ are dominant,
indicating that most of the Cl is in reservoirs. While inside the water plume, both the H2O_SO2
and H2O_SO2_ClO cases show strong depletion of the reservoirs HCl and $ClONO_2$, and most of
the Cly is either in the form of HOCl (a short-lived reservoir) or is activated in the form of ClO.
Unlike the chlorine activation process in the polar winter, HOCl is the highest in the HTHH
plume because heterogeneous chemistry is not fast enough to destroy HOCl to produce ClO. In
the case without heterogeneous chemistry, HCl and $ClONO_2$ are dominant in the plume,
indicating that heterogeneous chemistry is the main process of converting HCl to active chlorine.





Comparing total Cly and ClO in all panels, ClO does not exceed a quarter of the Cly, indicating
adding 0.00013Tg of ClO through injection is one way to produce the observed ClO. There is a
possibility that ClO is converted from other Cly species through chemical reactions we are not
aware of because this was a very unusual eruption.

(a)

(b)

(c)

**Figure 7.** Reactions inside and outside the plume in cm$^{-3}$sec$^{-1}$ and compound concentrations in
mol/mol. Red numbers represent values inside the plume, blue numbers outside the plume. **a)**
HOx balance and its interaction with Ox during daytime at 20 hPa on January 20, 2022. **b)**



Chlorine compound reactions in the H2O_SO2_ClO case. **c)** HOx cycle impact on $O_3$
production. Green arrows represent the heterogeneous reactions for chlorine activation. $H_2O$ is ~
600 ppm inside the plume and ~5.5 ppm outside the plume. Cly is ~ 4.2 ppbv inside the plume
and 1.5 ppbv outside the plume.

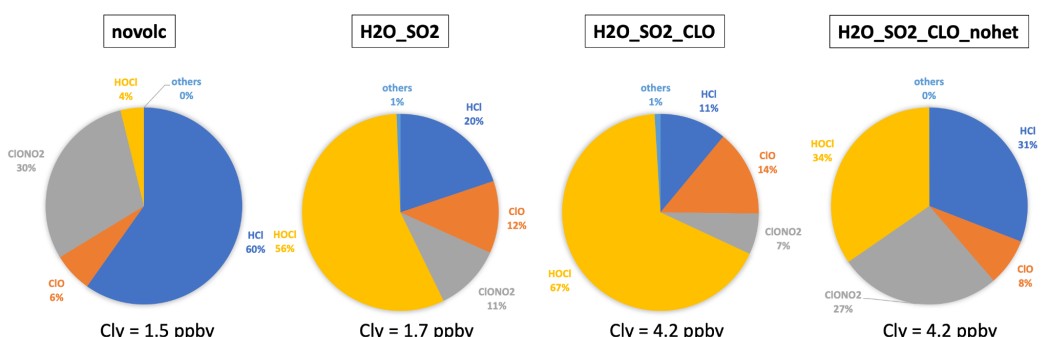

**Figure 8.** The percentage of each inorganic chlorine compound
($Cly=Cl+ClO+2Cl_2+2Cl_2O_2+OClO+HOCl+ClONO_2+HCl+BrCl$) inside and outside the plume.
The slight difference between novolc Cly and H2O_SO2 Cly is because $H_2O$ injection changes
the plume dynamics in the free-running simulations.

**4. Discussion**
The ozone loss inside the HTHH plume during the first ten days provides a unique
opportunity to study stratospheric chemistry and to understand the performance of the WACCM
state-of-the-art climate model, because the HTHH injected ClO and $H_2O$ exceed the normal
range of the stratospheric variability. These volcanic injections strongly altered the ClOx/HOx
interactions and heterogeneous reaction rates, producing different chemical pathways for
chlorine activation and ozone depletion from what occurs in the Antarctic ozone hole or Arctic
ozone depletion in the polar stratospheric winter and spring. HOCl is identified as playing a large
role in the in-plume chlorine balance and heterogeneous processes. The high HOCl
concentrations are a result of the very high in-plume water vapor content, which makes this event
different from chemistry in the Antarctic ozone hole, where $ClONO_2$ is more important.
This study also raises an interesting question of where the Cl comes from in the volcanic
injection. Seawater contains 3.5% sea salt, which implies that about 5 Tg of NaCl could have
been injected assuming that the injected 150 Tg of $H_2O$ came from sea water. However, we only
need to inject 0.00013 Tg of ClO to match the MLS ClO observations during the first few days
after the eruption. We also conducted a test injecting an equivalent amount of HCl (0.0009 Tg),
which resulted in a similar HOCl, ClO, and $O_3$ pattern (**Figure A2 and A3**). If we inject more
HCl or ClO, ClO would exceed the observed concentration, causing depletion of OH, and
slowing down the $SO_2$ oxidation. Evidently, if the water came from seawater, most NaCl was not
converted to HCl but stayed in the stratosphere as particles. *Vernier et al.* [2023] sampled NaCl
particles eight months after the eruption near Brazil. Based on their sampled NaCl concentration,
we estimate 0.5 to 1 Tg of NaCl may have been injected and stayed in the atmosphere. There are
several possibilities why this event did not inject 5 Tg of NaCl in the stratosphere: Remote
sensing particle size estimations [*Khaykin et al.*, 2022] and in-situ measurements [*Asher et al.*,
2023] indicates that the particles were submicron sized. However, sea salt particles injected into



the lower troposphere by wind are mainly particles larger than 10 µm. Hence, if the volcanic
injection had similar sized NaCl particles, most of them may have quickly fallen out of the
stratosphere. In addition, the majority of NaCl might have been washed out during the first
couple of hours of plume injection by acting as nuclei for ice particles. It is also possible that the
reactions that might release Cl from NaCl may not efficiently lead to reactive Cl. For example,
$HNO_3$ can react on sea salt heterogeneously very quickly in the troposphere to release HCl (De
Haan and Finlayson-Pitts, 1997; Guimbaud et al., 2002; Murphy et al., 2019). This reaction may
be accelerated by HTHH high humidity even if the temperature is low in the stratosphere. HCl
could be removed by condensing in supercooled water, which would reduce HCl vapor
concentrations by up to four orders of magnitude, preventing substantial stratospheric chlorine
injection [*Tabazadeh and Turco*, 1993]. Finally, it may be that the water injected came from
magmatic water, or seawater that percolated into the volcano and was released as steam. Such
water would not be rich in NaCl. In that case Cl observed by *Vernier et al.* [2023] may have been
bound up in minerals of the volcanic ash. Other halogen species such as bromine and iodine are
often observed after volcanic eruptions (large amounts of BrO were observed after HTHH in the
troposphere [*Li et al.,* 2023]). However, they can lead to much stronger ozone depletion if they
persist in the stratosphere. Since the elevated Cl in the model can well explain the $O_3$ depletion,
the impact of bromine and iodine on stratospheric $O_3$ is minimal for this eruption.
In addition, NOx can be produced by lighting inside or around the volcanic plume.
Observations show there was a record number of lightning events in this volcanic plume. Almost
400,000 flashes were observed by the GLD360 network over the 6 hours of the most active
eruption period (and ~590,000 total flashes) [Global Volcanism Program, 2022]. Considering
that tropospheric global models use a lightning source of 5 Tg(N)/yr and an average flash the
OTD/LIS satellite sensors produced an average global flash rate of 44±5 flashes per second, an
injection of N of ~0.001- 0.003 Tg (0.002 - 0.006 Tg of NO) would be expected for the HTHH
eruption. We conducted a model run with $H_2O$, $SO_2$, and an injection of 0.003 Tg of NO,
showing that this additional NO has little impact on the $O_3$ loss and ClO levels during the first
ten days (**Figure A6**). Therefore, lighting NOx probably does not contribute to the HTHH initial
in-plume $O_3$ loss. Because of the high water, NO would convert to $HNO_3$ in the first couple of
days. Unfortunately, we lack observations of $HNO_3$, NO, or $NO_2$ right after the eruption. MLS
observations in February (**Figure A7**) and the model simulations with $H_2O$ injection or $H_2O$+NO
injections show elevated $HNO_3$ compared with the background.
**Appendix A**

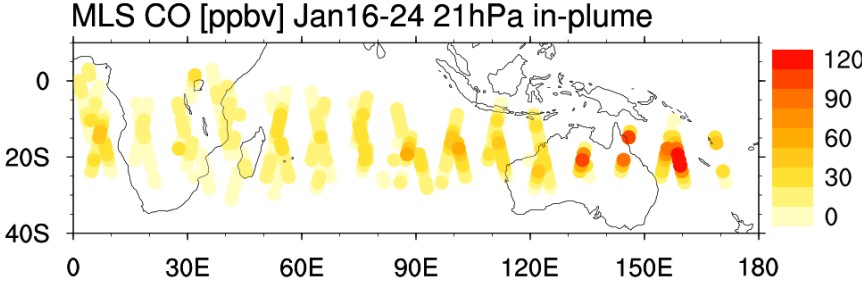

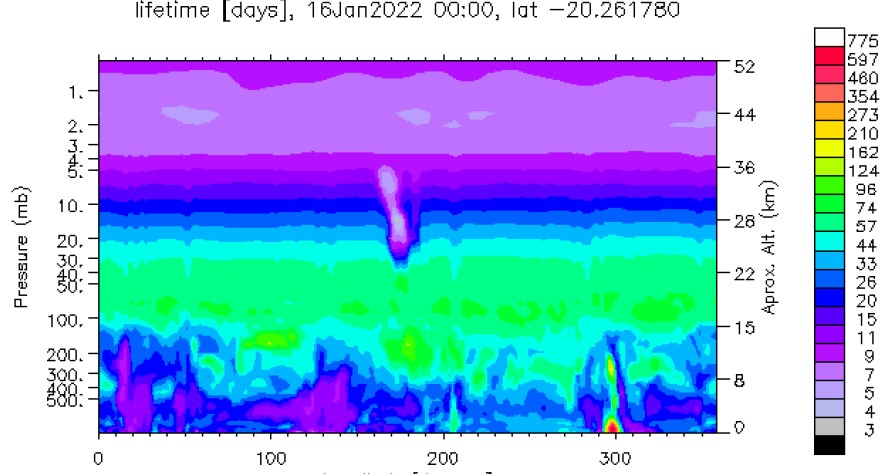

**Figure A1.** Top panel shows the MLS in-plume CO observation during the first 10 days after the
eruption. The bottom panel shows the CO lifetime on Jan 16 at 20°S is shortened from a month
to a few days because of the volcanic water plume. The observed CO mixing ratios of around
120 ppmv seem incompatible with typical CO levels over oceanic regions, indicating the
production of CO within the magma chamber or in the hot plume itself.


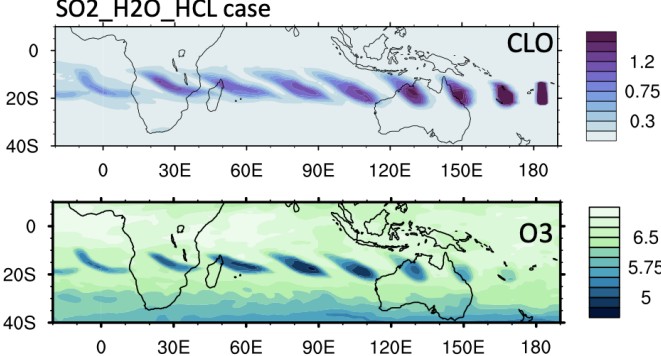

**Figure A2.** The $O_3$ and ClO evolution from the model case with an HCl injection of 0.000092
Tg (equivalent to 0.00013 Tg of ClO injection).




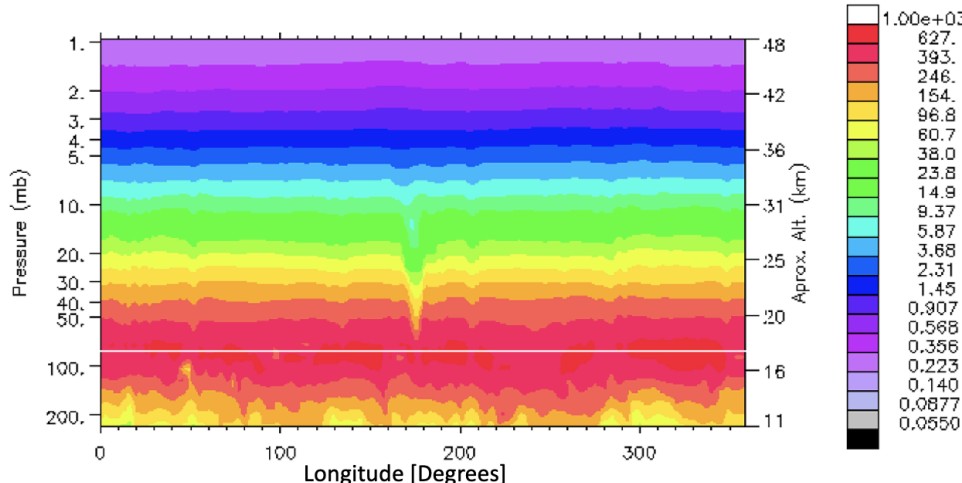

**Figure A3.** The HOCl evolution from the three model cases.
**Figure A4.** O$_3$ chemical lifetime is about 1 to 2 months at 20 hPa and is reduced to 10 days at the
HTHH location.




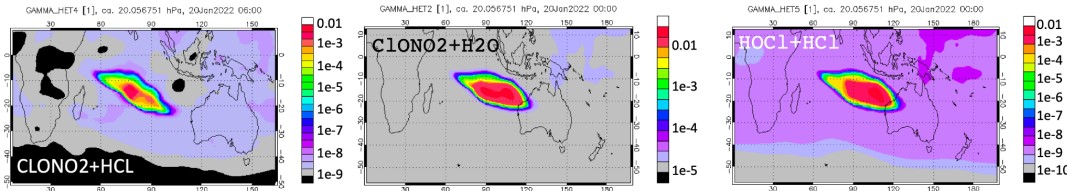

**Figure A5.** Heterogeneous reaction probabilities for the three heterogeneous reactions on January 20 at 20 hPa.

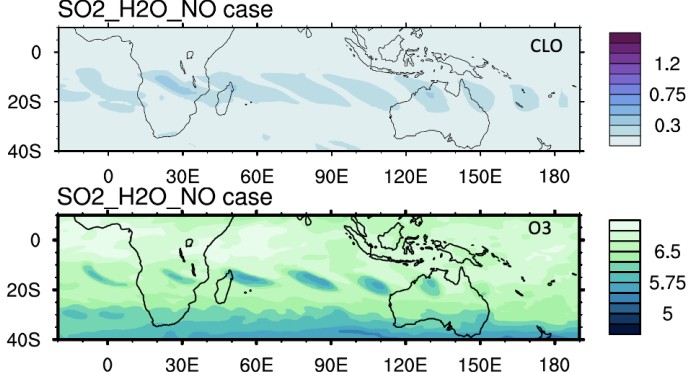

**Figure A6.** O$_3$ and ClO evolution from the model case with NO injection of 0.003 Tg, which is identical to the SO2_H2O case. The ClO and O$_3$ enhancement are due to the H$_2$O injection.

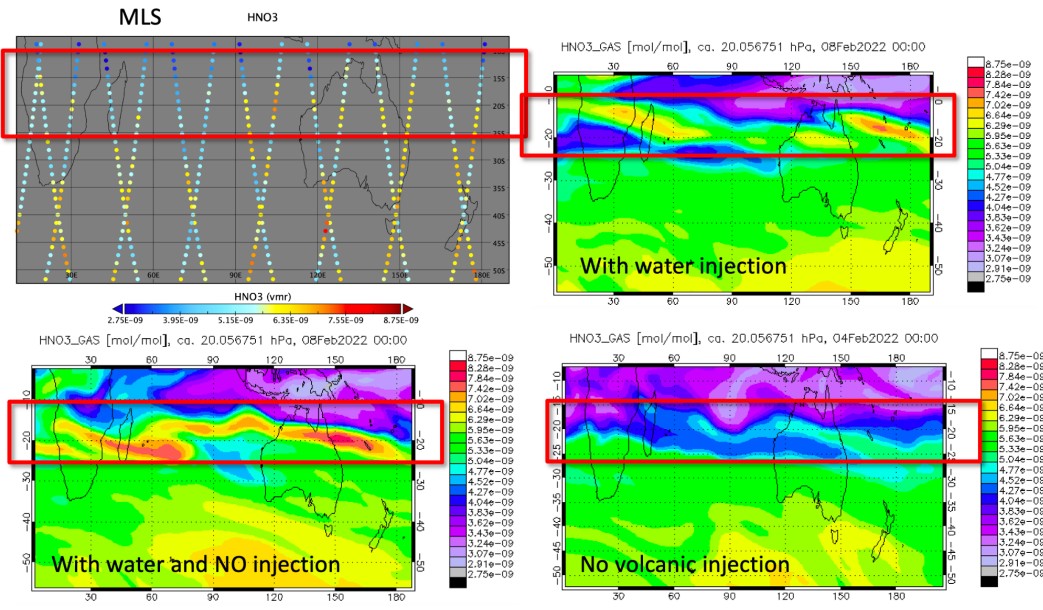



**Figure A7.** HNO$_3$ observed by MLS on February 8, 2022 compared to the model simulation with water and NO injection, as well as the no volcanic injection case. MLS shows similar elevated HNO$_3$ as the simulation case with H$_2$O injection or with H$_2$O/NO injection.

**Code availability:** The CESM2 model is available on the CESM trunk to any registered user at www.cesm.ucar.edu.

**Data availability:** The main simulation data generated during this study are available at (https://osf.io/f69ns/) with a permanent DOI 10.17605/OSF.IO/F69NS. Aura MLS v4 data is available at https://disc.gsfc.nasa.gov/datasets?page=1&keywords=AURA%20MLS. Water vapor radiosonde data is available at https://doi.org/10.5065/p328-z959 (26).

**Author contribution:** YZ, RWP, DK, and KHR designed the experiments and YZ performed the simulations. YZ prepared the manuscript with contributions from all co-authors. DK examined the sensitivity of the stratospheric H$_2$O abundance on the reaction probability (Figure 5). LM, HV and SE provided observational data and analysis. RWP, DK, OBT, JZ, ST, CGB, XW, WJR and KHR participated in the modeling data analysis.

**Competing interests:** At least one of the (co-)authors is a member of the editorial board of Atmospheric Chemistry and Physics.

**Acknowledgement**

This project received funding from NOAA's Earth Radiation Budget (ERB) Initiative (CPO #03-01-07-001). This research was supported in part by NOAA cooperative agreements NA17OAR4320101 and NA22OAR4320151. We thank Hazel Vernier, Dr. Kimberlee Dube, Dr. Pengfei Yu, Fracis Vitt, Dr. Ru-shan Gao, Dr. Margaret Tolbert, Dr. Micheal Mills, Dr. Daniel Murphy, and Dr. Brian Ridley for their valuable input. NCAR's Community Earth System Model project is supported primarily by the National Science Foundation. This material is based upon work supported by the National Center for Atmospheric Research, which is a major facility sponsored by the NSF under Cooperative Agreement No. 1852977. Computing and data storage resources, including the Cheyenne supercomputer (doi:10.5065/D6RX99HX), were provided by the Computational and Information Systems Laboratory (CISL) at NCAR. Work at the Jet Propulsion Laboratory, California Institute of Technology, was carried out under a contract with the National Aeronautics and Space Administration (80NM0018D0004).

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
