# Peer review of "Stratospheric ozone depletion inside the volcanic plume shortly after the 2022 Hunga Tonga eruption"

_EGUsphere, 2023_

## Author Response (AR1)

**Review #1**

We appreciate your time and effort to review the manuscript. Your thoughtful comments helped us to improve the quality of the manuscript. Please view our detailed response below for each question. The question is noted in black, and our response in blue.

**General comments**

The paper analyses stratospheric ozone depletion in the plume of a strong volcanic eruption with high water vapor and chlorine emissions, using satellite and radiosonde data and a comprehensive chemistry climate model. In general it is well written, provides the most important references (including classical ones) and should be published after some minor improvements.

**Specific comments**

Line 69: The full name of the model should be mentioned here, line 96 is too late.
Fixed.

Line 98: The horizontal resolution of the model should be provided too. How many grid points are typically within the volcanic plume?
Line 104, Add "The model has a horizontal resolution of 0.9° latitude × 1.25° longitude. The injection plume in the model includes about 40 grid points."

Line 178 or earlier: Here one or two references should be provided. Line 364 is too late.
Line 182, add "[Shi et al., 2001]".

Line 209: Refer to Fig.2 concerning the time (As .. but ..).
Line 147: add "from January 18 - 24"
Line 231: add "from January 16 - 24"
Line 231: add ""In-plume" is defined as in Figure 2. Note that MLS ozone retrievals were unaffected by the plume leading to the addition of two extra days of data for this figure."

Line 276: I suppose this is due to the high HOCl concentration and not due to reduced overhead ozone since the plume is at different locations at different altitudes due to the winds. Please expand a little bit here.
We checked the HOCL photolysis reaction rate in the model. The fractional change of HOCl is almost the same as the fractional change of HOCl j value.
We revised the Figure 7b, which ignored the small value of HOCl before.
Line 307 add "due to the high HOCl mixing ratio".

Line 291ff: Is this the NO scenario of Table 1 or is that only in the discussion section? Please clarify.
Line 324: add "Note that we don't inject lightning NOx in this case, a possible scenario during the eruption phase, that can also further increase the $O_3$ production (detailed in the discussion

section)."

Line 381f: This can cause also ozone production involving $HO_2$ and $CH_3O_2$.
CH3O2+NO is also twice as much as before. But much smaller than NO+HO2. So we haven't put it in the text.
Line 395: Add "Compared to the H2O_SO2 case, the simulated $O_3$ loss in the H2O_SO2_NO case increased by $\sim 5x10^5$ molecules/cm$^3$/sec, but at the same time, the $O_3$ production rate increased by $\sim 5x10^5$ molecules/cm$^3$/sec. The NO+HO$_2$ reaction rate in the H2O_SO2_NO case increases 5 times compared with the H2O_SO2 case."

**Technical corrections**

Please use a larger font in Fig.8.
Fixed.

**Review #2**

We appreciate your time and effort to review the manuscript. Your thoughtful comments helped us to improve the quality of the manuscript. Please view our detailed response below for each question. The question is noted in black, and our response in blue.

This is a nice model simulation analysis study trying to detangle the impact of various chemical compounds injected from HTHH eruption on stratospheric ozone depletion. The set of simulations are carefully designed and the paper is well written and should be accepted for publication discussion paper on ACPD after some minor revision.

1. "lightning" should be capitalized.
   Fixed.

2. I guess by you are implying short-term ozone depletion by using the term "in-plume", but this might look obvious to readers. I would suggest to add something like "near-term in-plume" or "after eruption in-plume" to let readers know that you are only looking at the first 10 days impact.
   Line 23: change to "Near-term in-plume ozone depletion".

3. L98-99. The correct reference should be "GEOS-5 MERRA-2 meteorological reanalysis".
   Fixed.

4. L157-L176. I am not a big fan of people starting the discussions/sentences with "Figure 3a shows …". A punchier way would be start with a scientific statement sentence with figure references in parenthesis, as you did in L193-L206. This issue is more problematic in this paragraph, as a lot of the sentences here are figure description and belongs in the figure caption, not in the text. The way it is written now, constantly switching between figure description and results interpretation, is very distracting. I would suggest re-write, following L193-L206 as a style example.
   rewrote this part Line 168-180: "MLS observed in-plume low ozone concentration at 20 hPa (**Figure 3a**), especially during these three days: ozone concentrations of 4.8 ppmv on January 17, 4.6 ppmv on January 20, and 5.1 ppmv on January 24. These are ozone anomalies of about 1.7 ppmv, 1.9 ppmv, and 1.4 ppmv, respectively. The anomalies are calculated using the background average values in this area (6.5 ppmv) subtracting the low ozone values. Note that any interpretation of these $O_3$ anomalies needs to consider the coarse MLS vertical resolution (~3 km). Because the plume is spatially small during the initial days, MLS tracks do not capture the maximum plume perturbation every day. The simulation with the water injection (**Figure 3b**) accelerates the HOx catalytic cycle and shows evident $O_3$ reduction, but less than observed. Once we inject ClO on top of the massive water injection (**Figure 3c**), $O_3$ loss is significantly enhanced and is close to the observations after January 18. The difference between **Figure 3d** and **Figure 3c** is caused by heterogeneous reactions, which usually only happen in the stratospheric polar springtime where they cause the Antarctic ozone hole and Arctic ozone depletion."
   Move several sentences to Figure 3 caption "**Figure 3d** uses the same injection as **Figure**

**3c** but with heterogeneous reactions (i.e., HCl+HOCl, ClONO$_2$+H$_2$O, and ClONO$_2$+HCl) turned off. The simulated O$_3$ in the H2O_SO2 case uses one model time step each day that occurs near local noon."

5. L273-L290. Can you be more qualitatively descriptive in rate changes, instead of just stating the numerical values? For example, instead of saying HO2+O3 reaction rate increases from 5x10^4 to 5x10^5 cm-3 sec-1, you can say the reaction rate increased by a factor of ten. Easier to comprehend if described that way.
Fixed from L302 to L326.

6. L291-L297. See comment #4 above.
Line 321: Change the sentence to "Besides the ozone loss reactions, ozone production reactions are also significantly altered by the water plume (**Figure 7c**)."
Line 327: change the sentence to "Comparing the partitioning of Cly (Cl+ClO+2Cl$_2$+2Cl$_2$O$_2$+OClO+HOCl+ClONO$_2$+HCl+BrCl) reveals the in-plume chlorine activation processes (**Figure 8**)."

7. L330-333. I like this sentence, very good motivation description. May be add something similar in Introduction as well?
rewrite line 69-74: "*Evan et al.* [2023] report observations of decreased O$_3$ and HCl, and increased ClO in the first week following the HTHH eruption at 20 hPa, which is related to the injected H$_2$O exceeding the normal range of the stratospheric variability. Here we use the Whole Atmosphere Community Climate Model version 6 (WACCM6) model [*Zhu et al.*, 2022] to analyze the dynamic and chemical contributors to this initial in-plume ozone depletion, and to understand the climate model performance."